# Debiasing Multimodal Models via Causal Information Minimization

**Vaidehi Patil**     **Adyasha Maharana**     **Mohit Bansal**
UNC Chapel Hill
{vaidehi, adyasha, mbansal}@cs.unc.edu

## Abstract

Most existing debiasing methods for multimodal models, including causal intervention and inference methods, utilize approximate heuristics to represent the biases, such as shallow features from early stages of training or unimodal features for multimodal tasks like VQA, etc., which may not be accurate. In this paper, we study bias arising from confounders in a causal graph for multimodal data, and examine a novel approach that leverages causally-motivated information minimization to learn the confounder representations. Robust predictive features contain diverse information that helps a model generalize to out-of-distribution data. Hence, minimizing the information content of features obtained from a pretrained biased model helps learn the simplest predictive features that capture the underlying data distribution. We treat these features as confounder representations and use them via methods motivated by causal theory to remove bias from models. We find that the learned confounder representations indeed capture dataset biases and the proposed debiasing methods improve out-of-distribution (OOD) performance on multiple multimodal datasets without sacrificing in-distribution performance. Additionally, we introduce a novel metric to quantify the sufficiency of spurious features in models' predictions that further demonstrates the effectiveness of our proposed methods.[1]

## 1 Introduction

The success of multimodal models in various tasks has been attributed to their ability to rely on spurious correlations (or biases) present in the training data (Jabri et al., 2016; Agrawal et al., 2016; Zhang et al., 2016a; Goyal et al., 2017). An example of image bias in VQA is when the model tends to look at prominent objects in the image rather than focusing on the object about which the question

---

[1]Our code is available at: https://github.com/Vaidehi99/CausalInfoMin

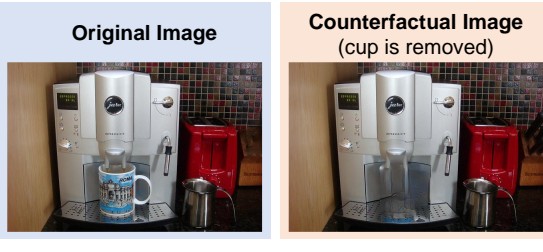

Figure 1: Multimodal models tend to rely on spurious correlations in the dataset to answer a question. Existing methods remove unimodal biases whereas our method removes biases arising from cross-modal interactions as well and is more invariant to irrelevant features (e.g., the coffee mug) in this example.

is asked (Wen et al., 2021) (see Fig. 1). These models leverage such biases to perform well on in-distribution (ID) evaluation data (Agrawal et al., 2018a). However, their poor performance on out-of-distribution data reveals that they merely rely on superficial features rather than capturing the true causal relationships between inputs and targets.

Existing methods attempt to diminish a model's reliance on these shortcuts by taking one or both of two primary strategies: (a) by balancing the sample groups with and without spurious correlation, e.g. via data augmentation (Gokhale et al., 2020) or sample synthesis (Chen et al., 2020, 2022; Kolling et al., 2022a), and (b) by explicitly eliminating the impact of spurious correlations during model training or inference (Huang et al., 2022; Lin et al., 2022; Pan et al., 2022). In the former approach, the identification of the unique set of spurious correlations in each sample becomes essential to curate augmented samples for achieving balance. Consequently, approaches that alleviate biases in features or predictions, independent of the availability of non-spurious data, are more desirable. Such meth-

ods also offer the additional advantage of being agnostic to the specific dataset and task at hand.

Recent research on debiasing models has emphasized the significance of causal theory (Zhang et al., 2021; Liu et al., 2022; Bahadori and Heckerman, 2020) i.e., many spurious correlations originate from confounding variables that induce non-causal dependencies between inputs and labels (Pearl et al., 2000). However, effectively identifying and representing biases that undermine prediction accuracy remains a challenging task. Previous studies on multimodal models have utilized image features from early training stages as contextual biases for multi-label image classification (Liu et al., 2022), or introduced unimodal training branches to mitigate spurious correlations in Visual Question Answering (VQA) (Niu et al., 2021). Moreover, these approaches overlook biases stemming from multimodal interactions within their causal graphs. Hence, in this work, we represent the bias as confounder variables that have a direct causal effect on multimodal features and the corresponding predictions (see Fig. 2(a)). Spurious correlations represent the simplest predictive features that explain biased datasets (Geirhos et al., 2020), thereby making them easily learnable by machine learning models under limited representation capacity (Yang et al., 2022). We capitalize on this notion to study a novel framework that combines information theory and causal graphs to learn confounder representations capable of capturing spurious features. We examine two approaches to learn the confounder representations by imposing information loss on biased multimodal features i.e., (a) *latent variable modeling* using a generative model and (b) *rate-distortion* minimization (Shannon, 1948). Subsequently, we utilize these confounders in our proposed debiasing methods, namely ATE-D and TE-D, leveraging the concepts of *average treatment effect* (Glymour et al., 2016) and *total effect* (Pearl, 2022) causal mechanisms, respectively.

In ATE-D, we employ an autoencoder to reconstruct the biased features. The autoencoder projects these features into a lower-dimensional latent space, capturing latent features that act as substitutes for unobserved confounders (Huang et al., 2022). By clustering the learned confounder representations across the dataset, we construct a dictionary of confounders. We subsequently perform backdoor adjustment based on the average treatment effect, utilizing feature reweighting (Kirichenko et al.,

2022). In TE-D, we leverage the *rate-distortion function* which controls the number of bits required to encode a set of vector representations (Chowdhury and Chaturvedi, 2022). We minimize the rate-distortion function for a non-linear projection of the features extracted from a biased pretrained model, while simultaneously minimizing the cross-entropy loss of predicting from these projected features. This results in the loss of diverse information from the features and the retention of simple features that are also maximally predictive of the biased dataset. We treat these features as the confounder representations that stem from spurious correlations in the dataset and compute the (unbiased) *total effect* of the input by taking the difference between the biased feature and its respective confounder.

We evaluate the proposed methods on several multimodal tasks and along multiple dimensions i.e., in-distribution and out-of-distribution performance, efficiency, and robustness. Results show that these methods not only outperform baseline models with lower training overhead but also yield additional gains on top of unimodal debiasing methods. In this work, we demonstrate the presence of multimodal biases and the need for multimodal debiasing along with the potential of confounder modeling via information loss in causal multimodal debiasing. Our contributions are as follows:

- We present two methods, TE-D and ATE-D, that leverage causally-motivated information loss to learn confounder representations from biased features and utilize them to debias models.

- Our methods remove multimodal biases and yield up to 2.2% and 2.5% gains over LXMERT (Tan and Bansal, 2019), on VQA-CP and GQA-OOD (Kervadec et al., 2021) datasets respectively, and 0.7% gains on top of unimodal debiasing (Wen et al., 2021). Importantly, our methods exhibit superior parameter efficiency and reduced training time compared to existing debiasing methods.

- We propose a sufficiency score ($\lambda$) for quantifying the reliance of models on spurious features. Results show that our methods improve robustness to spurious correlations in the dataset.

- We analyze the confounders learnt in ATE-D, TE-D and show that they encode dataset biases.

## 2 Related Work

**Data Augmentation.** Balancing data (Zhang et al., 2016b) can involve training a generative

model for sample synthesis (Agarwal et al., 2020; Sauer and Geiger, 2020), designing suitable data selection heuristics (Chen et al., 2020), or curating balanced/counterfactual samples (Goyal et al., 2017; Gokhale et al., 2020; Kolling et al., 2022c). Human explanations can be used as additional training signals to promote reasoning (Ying et al., 2022; Wu and Mooney, 2019; Selvaraju et al., 2019). We debias models using existing biased data.

**Inductive Bias in Model Architecture.** Agrawal et al. (2018a) explicitly design inductive biases to prevent the model from relying on training priors. Clark et al. (2019); Cadene et al. (2019); Ramakrishnan et al. (2018) rely on a separate QA branch to weaken the language prior in VQA models via adversarial or multi-task learning. Wen et al. (2021) use contrastive loss to remove unimodal biases for VQA. Peyrard et al. (2022) discover invariant correlations in data across different training distributions to enable generalization.

**Inductive Bias for Modeling Confounders.** Kallus et al. (2018) recover latent confounders via low-rank matrix factorization and Sen et al. (2017) utilize low-dimensional variables for encoding confounders. We use low-dimensional features to limit representational capacity for encoding confounders in multimodal data.

**Causal Perspective.** Lin et al. (2022) use causal intervention through backdoor adjustment (Glymour et al., 2016) to disentangle the biases for unsupervised salient object detection. Huang et al. (2022) use ATE to debias referring expression models. Niu et al. (2021) compute the Total Indirect Effect (TIE) of the multimodal branch to omit the influence of unimodal branches. Veitch et al. (2021) formalize counterfactual invariance and its relation to OOD performance. Liu et al. (2022) use features from early training as confounders and compute the Total Direct Effect (TDE) for multi-label image classification. We combine information theory and causal theory to learn confounders from biased representations and use them via ATE and TE causal mechanisms to debias a model.

## 3 Causal Theory Preliminaries

In this section, we discuss our proposed causal graph for multimodal tasks and the two causal mechanisms relevant to our debiasing methods.

**Causal Graph.** Causal graphs are directed acyclic graphs $\mathcal{G} = \{\mathcal{V}, \mathcal{E}\}$ where the edges $\mathcal{E}$

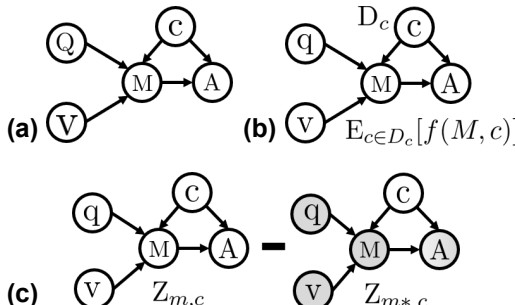

Figure 2: Demonstration of (a) our proposed causal graph for multimodal tasks, (b) Average Treatment Effect (ATE), and (c) Total Effect (TE) on (a). Values in grey indicate the 'no-treatment' condition.

are used to represent causal relationships between random variables $\mathcal{V}$. When the variable $\mathbf{Q}$ has an *indirect effect* on $\mathbf{A}$ through a variable $\mathbf{M}$ i.e. $\mathbf{Q} \rightarrow \mathbf{M} \rightarrow \mathbf{A}$, the variable $\mathbf{M}$ is said to be a *mediator* in the causal graph (see Fig. 2(a)). If a variable $\mathbf{C}$ has a *direct causal effect* on both $\mathbf{M}$ and $\mathbf{A}$, it is said to be a *confounder*.

**Causal Perspective for Multimodal Tasks.** Multimodal models for tasks combining vision ($V$) and language ($Q$) often face the challenge of confounding variables, which introduce spurious features. Current approaches rooted in causal theory aim to mitigate direct unimodal effects. However, a VQA example (Fig. 1) highlights a limitation: models trained predominantly on centrally located objects struggle with queries about obscured object colors. Existing causal graphs for multimodal tasks fail to account for spurious correlations arising from such interactions. To address this, we propose a confounder $\mathbf{C}$ that influences both the mediator $\mathbf{M}$ and the answer $\mathbf{A}$ (Fig.2(a)). By modeling biases encoded in multimodal features as confounder $\mathbf{C}$, we can eliminate biases using causal intervention.

In order to debias VQA models, we adopt two causal mechanisms i.e., the Average Treatment Effect (ATE) and Total Effect (TE), which essentially refer to the same quantity but differ in how they deal with the confounder (VanderWeele, 2015; Tang et al., 2020a). In ATE, $C$ is treated as a distribution, and $c$ is sampled by assuming implicit causal association with the treatment $M = m$. In TE, $c$ has an explicit causal association with the treatment $M = m$ in each sample. We explore both in our work and discuss their theories below.

**Average Treatment Effect.** The aim of causal inference is to estimate the independent effect of an intervention on a treatment variable $M$ on

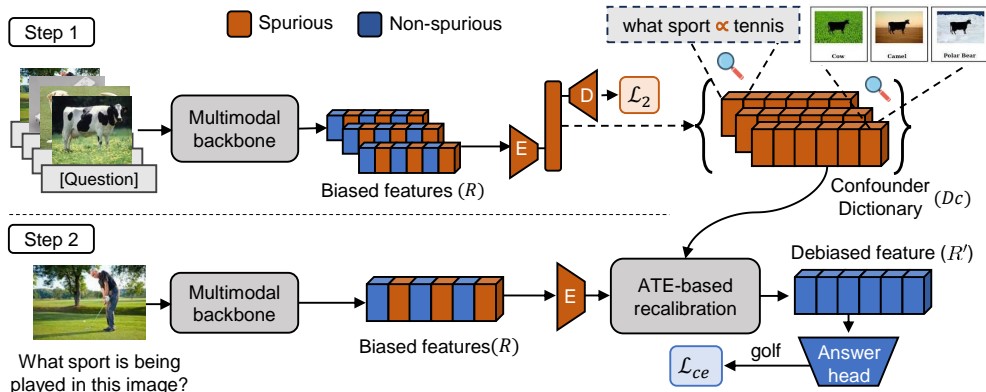

Figure 3: An illustration of our method ATE-D based on autoencoder-based confounder modeling and Average Treatment Effect causal mechanism (see Sec. 4.1). The confounders are modeled using autoencoder in Step 1 and biased features are recalibrated using confounders to get debiased features in Step 2.

an outcome of interest $A$ i.e. to estimate the conditional probability distribution $P(A|do(M))$ where the *do*-operation implies the causal effect of $M \rightarrow A$. However, standard models are optimized to infer the observational conditional probability $P(A|M)$. In the presence of confounders i.e. variables $c \in C$ that affect both $A$ and $M$, $P(A|M) \neq P(A|do(M))$. $P(A|do(M))$ can be estimated using backdoor adjustment by controlling for all values of the confounders $c \in C$ as:

$$P(A|do(M)) = E_{c \sim C}[P(A|M, c)] \quad (1)$$

This translates to an empirical sum over all possible values of the confounder in practice, also known as the average treatment effect (ATE) (see Fig. 2(b)). When the confounders are known and observed, the confounder values are selected using suitable heuristics (Pearl et al., 2000). However, observing all confounders is not always possible. Hence, we model the variables that can be used as substitutes for the confounders via latent representations in autoencoders (Sen et al., 2017; Kallus et al., 2018). Huang et al. (2022) use average treatment effect-based debiasing for the task of visual grounding by modeling confounders.

**Total Effect.** We need to isolate the causal effect of $M = m$ on $A$, free from the influence of the confounders $C$. According to causal theory, the total effect (TE) of treatment $M = m$ on $A$ is,

$$TE = A_{m, C_m} - A_{m*, C_m} \quad (2)$$

where $M = m$, $M = m*$ represent 'treatment' and 'no treatment' conditions respectively; $C_m$ is the confounder under treatment and $A_{m, C_m}$ is the answer in the presence of treatment as well as con-

founder. The direct effect of $C_m$ on $M$ is eliminated by retaining the confounder on both sides of the difference (see Fig. 2(c)). In practice, we take the difference between feature representations of $A_{m, C_m}$, $A_{m*, C_m}$ i.e. $Z_{m,c}$, $Z_{m*,c}$ respectively, to eliminate the effect of $C_m$ (see Sec. 4.2).

# 4 Debiasing Methods: ATE-D and TE-D

Kirichenko et al. (2022) show that machine-learning models learn spurious as well as non-spurious features when trained on a biased dataset, but over-rely on the former for making predictions. In Sec. 1, we discussed how confounder variables contribute to these spurious predictions. Further, Yang et al. (2022) show empirically that deep models preferentially encode dataset shortcuts under limited representation capacity. Indeed, neural nets are expected to trade-off between maximal compression of the learnt representations and maximal fitting to the labels (Information-Bottleneck) (Shwartz-Ziv and Tishby, 2022). Hence, we propose information minimization, by limiting representation capacity via low-dimensional vectors, to learn the bias/confounder features. Similar approaches exist i.e. Kallus et al. (2018) recover latent confounders by performing low-rank matrix factorization on high-dimensional data, and Sen et al. (2017) use low-dimensional variable to encode confounder. We propose two methods to learn and use confounder features for debiasing: (a) latent variable modeling in ATE-D and (b) rate-distortion minimization in TE-D. In both approaches, the biased features are projected into low-dimensional vectors through various mechanisms, limiting their representation capacity and promoting information minimization. Subsequent

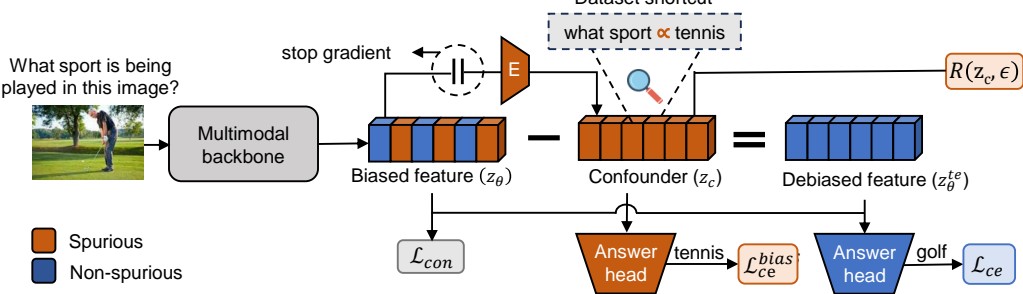

Figure 4: An illustration of our method TE-D based on Rate-Distortion & Total Effect causal mechanism (see Sec. 4.2). The biased features are used to learn confounder features guided by rate-distortion minimization and cross-entropy loss ($L_{ce}$). The confounders are subtracted from the biased features to get debiased features.

sections further elaborate these methods.

## 4.1 ATE-D: Deconfounding Using Average Treatment Effect

We follow a 2-step framework where we start with a pre-trained biased model, then (1) obtain the substitute confounders from the latent variables of autoencoder (Huang et al., 2022) and (2) use these confounders to debias the pretrained model using feature reweighing (Kirichenko et al., 2022).

**Step 1:** We collect the biased features $r \in R$ from a biased model for all samples in the training data and train an autoencoder composed of dense layers ($F_{enc}, F_{dec}$) to encode them into a lower dimension (see top, Fig. 3). The latent dimensions of the generative model capture the most common biases in the dataset and serve as a substitute for the confounders. We use a small-capacity network in order to capture the biases stemming from spurious correlations in the latent dimensions and avoid encoding the correct predictive features. $F_{enc}, F_{dec}$ are trained using the reconstruction loss $L_{recon} = d(R, R)$, where $d(,)$ is the Euclidean distance function. We model the substitute confounders $\hat{c} \in \hat{C}$ for R ($\hat{\cdot}$ represents approximation) and cluster them to get a dictionary $D_{\hat{c}}$, which represents the main elements of $\hat{C}$ for efficient backdoor adjustment (Eqn. 1).

**Step 2:** Kirichenko et al. (2022) show that nonspurious features can be emphasized in biased features by reweighing them using a balanced dataset. However, creating balanced data is nontrivial for complex tasks like VQA. To overcome this challenge, we instead create an instantiation of backdoor adjustment that reweighs biased features based on their similarity with the substitute confounders (see bottom, Fig. 3). We hypothesize that this leads to lower weights for the simple spuri-

ous features and higher weights for more complex predictive features, alleviating the over-reliance on spurious features for prediction. For a sequence of biased features $r = [r_1, r_2, ..., r_k]$, we recalibrate each $r_i$ according to their similarity with the confounders in $D_c$ i.e., the weight $w_i$ for $r_i$ is,

$$w_i = 1 - \frac{1}{len(D_{\hat{g}})} \sum_{g_j \in D_{\hat{g}}} s(F_{enc}(r_i), g_j) \quad (3)$$

where $s(.)$ is the cosine-similarity function (see ATE-based recalibration in Fig. 3 and see Appendix for explanation of recalibration as an instantiation of back-door adjustment).

$$r'_i = w_i * r_i; R' = [r'_1, r'_2, ..., r'_k] \quad (4)$$

The resulting debiased features $R'$ are then used to replace $R$ as shown in Fig. 3.

## 4.2 TE-D: Debiasing Using Rate-Distortion & Total Effect

The rate-distortion function $R(Z, \epsilon)$ measures the minimum number of bits per vector required to encode the sequence $Z = \{z_1, z_2, ...z_n\} \in \mathcal{R}^{n \times d}$ such that the decoded vectors $\{\hat{z}\}_{i=1}^n$ can be recovered up to a precision $\epsilon^2$ i.e.,

$$R(Z, \epsilon) = \frac{1}{2}\log_2 \det(I + \frac{d}{n\epsilon^2}ZZ^T) \quad (5)$$

where $\frac{1}{n}ZZ^T$ is the estimate of covariance matrix for the Gaussian distribution (Chowdhury and Chaturvedi, 2022) and assuming that the vectors are i.i.d. samples from $\mathcal{N}(0, 1)$. Rate-distortion values are higher for distribution with high variance (diverse features). Hence, we minimize the rate-distortion to learn confounder representations in TE-D. Our implementation is illustrated in Fig. 4. Given a *biased model* with parameters $\theta$, we first obtain the biased feature $z_\theta$. Then, we encode the

$z_\theta$ into a lower dimension to promote information loss, along with a classification head ($\mathcal{L}_{ce}^{conf}$) to encourage retaining predictiveness of the information present in the encodings, which we treat as the confounder representation $z_c$. Finally, we enforce rate-distortion minimization ($R(z_c, \epsilon)$) on $z_c$ for promoting the loss of complex feature information. We enforce a stop gradient (see in Fig. 4) prior to the encoder in order to prevent the training signals for learning confounder representations from seeping into the parameters of the biased model.

In order to isolate the causal effect of $M$, we need to cut off the link $C \rightarrow M$ (see Fig. 2(c)). This can be achieved by computing the total effect (see Sec. 3) i.e., $A_{m,c} - A_{m*,c}$, where $m$ and $m*$ represent the treatment and no-treatment conditions respectively, while $c$ represents the confounder resulting from $M = m$. We implement this at the feature level by representing $A_{m,c}$ with the biased features $z_\theta$ and $A_{m*,c}$ with the confounder features $z_c$. Next, we take the difference of those features to secure $z_\theta^{te}$ which represents the direct effect of $M$. i.e. $z_\theta^{te} = z_\theta - z_c$. We further aid the debiasing process by enforcing a contrastive loss between the three sets of features $z_\theta, z_c, z_\theta^{te}$ as:

$$\mathcal{L}_{con} = \log \frac{\mathbf{e}^{s(z_\theta^{te}, z_\theta)}}{\mathbf{e}^{s(z_\theta^{te}, z_\theta)} + \mathbf{e}^{s(z_\theta^{te}, z_c)}} \qquad (6)$$

where $s(.)$ is the cosine similarity function. The contrastive loss penalizes the model when the confounder is correlated with the biased feature $z_\theta$ and hence, promotes debiasing of the multimodal backbone itself. In summary, we jointly optimize the model for learning confounder representations via $\mathcal{L}_{ce}^{conf}, R(Z_c, \epsilon)$ and debiasing with the help of the learned confounders via $\mathcal{L}_{con}, \mathcal{L}_{ce}$ i.e., $\theta_{deconf} = \text{argmin}_\theta \mathcal{L}_{con} + \mathcal{L}_{ce} + \mathcal{L}_{ce}^{conf} + \alpha R(Z_c, \epsilon)$, where $\alpha$ is the weight factor for rate-distortion loss.

### 4.3 Causal Debiasing vs. Data Augmentation

Data augmentation is an effective and popular method for enhancing model robustness (Puli et al., 2023; Gokhale et al., 2020; Chen et al., 2020), however, it presents certain constraints, particularly when employed in the context of debiasing within VQA models, such as:

**Dependency on prior knowledge.** Data augmentation typically hinges on pre-existing knowledge of potential biases within the dataset. For instance, Mikołajczyk-Bareła (2023) use knowledge of biases i.e. the presence of shape and texture bias

in data to augment data based on style transfer, Gokhale et al. (2020) identify unimodal biases to augment multimodal datasets. However, such awareness may not be comprehensive or entirely precise. Consequently, the efficacy of data augmentation is contingent on the accuracy and completeness of the a priori understanding of the biases underpinning the augmentation strategy. Conversely, methods that manipulate representation vectors directly to remove biases, such as our proposed debiasing techniques, extract spurious correlations from the data without requiring predefined assumptions about specific biases.

**Scalability and cost implications.** The creation of augmented datasets is often time-intensive as well as cost-intensive (Sauer and Geiger, 2020). The process demands domain expertise to adeptly identify and apply augmentations (Tang et al., 2020b). This resource-intensive nature of data augmentation can curtail its applicability, especially when used for models that must adapt to a multitude of diverse, evolving sources of bias.

Automated discovery of spurious correlations, as performed in our proposed methods ATE-D and TE-D, is advantageous over data augmentation when dataset biases are inadequately defined or in a state of perpetual flux. For instance, in numerous *real-world applications*, the dataset may harbor concealed or subtle biases that evade detection through manual inspection or domain expertise. Similarly, in *dynamic environments*, dataset biases can undergo periodic shifts. As a result, pre-established augmentation strategies become unviable for such scenarios. The techniques proposed in this work can adapt to the changing characteristics of data within a black box, making them more useful.

Another research thread aims to uncover coherent data subsets on which machine learning models exhibit subpar performance, such as the approach introduced in Domino (Eyuboglu et al., 2021). When these underperforming slices are accurately identified and labeled, it offers an opportunity to enhance model robustness by either updating the training dataset or employing optimization techniques designed to handle systematic performance issues in these slices. While this method aligns with our objective of improving the identification of systematic biases, slice discovery approaches achieve

it from a data perspective and require ground truth labels, whereas we take a distinct feature-based approach that does not rely on the ground truth.

## 5 Measuring Sufficiency & Necessity of Spurious Features in Multimodal Tasks

OOD generalization accuracies indicate the model's ability to learn causal relationships between inputs and labels (Veitch et al., 2021). Another approach to assess causal learning is by examining the models' invariance to spurious features in the dataset. Joshi et al. (2022) categorize spurious features into (a) *Type 1 Features* that are neither necessary nor sufficient for predicting the label e.g., 'person' (visual feature) when the VQA question is "How many trees are in the picture?" (see left, Fig. 5) (b) *Type 2 Features* that are necessary but not sufficient to make predictions e.g., the feature "Is the man" (see right, Fig. 5). When a model consistently answers "yes" to all "Is the man..." questions regardless of the image, it is considered to exhibit spurious behavior. We employ this framework to analyze debiasing methods in our experiments.

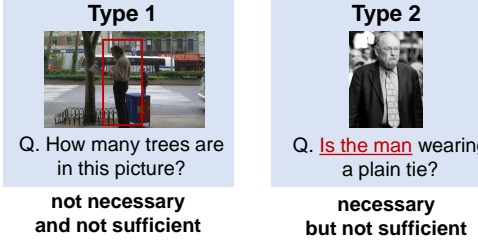

Figure 5: Types of spurious features (red) in VQA based on necessity and sufficiency.

**Necessity.** To assess the robustness of models to Type 1 features, we compare their performance on samples with and without a specific Type 1 feature. In an unbiased model, the absence of this feature should have no impact on performance. However, a biased model tends to rely on it due to spurious correlations that confound the features and labels. An effective debiasing method should render the model invariant to such features. Type 1 features predominantly arise from the image in multimodal tasks, as depicted in Fig. 5. Therefore, we evaluate the necessity of these features using counterfactual images (Agarwal et al., 2020) (refer to Sec.6).

**Sufficiency.** To assess the robustness of models to Type 2 features, we propose a new metric for measuring the sufficiency of a feature in relation

to a prediction. The certainty of predictions is determined by the Kullback-Leibler (KL) divergence between the predicted output distribution and a uniform distribution across all samples in the group (Ying et al., 2022). We define the sufficiency score ($\lambda$) as the percentage of the model's certainty that can be attributed to the spurious component of the input in making a prediction. For a data sample $(x, y)$, where the input $x$ consists of the spurious feature $x^s$ and the remaining context $x^c$, i.e., $x = [x^s; x^c]$, we compute the sufficiency $\lambda$ as:

$$\lambda = \frac{\sum_{i=1}^{G} \text{KL}(f(y_i|x_i^s)||\mathbf{U})}{\sum_{i=1}^{G} \text{KL}(f(y_i|x_i)||\mathbf{U})} \quad (7)$$

Here, $\mathbf{U}(.)$ represents the uniform distribution, $f(.)$ denotes the trained model, and $G$ is a group of samples. A reliable debiasing technique should reduce the sufficiency of spurious features. In the case of the multimodal Visual Question Answering (VQA) task, where $x_i = (q_i, v_i)$, we evaluate sufficiency of Type 2 features that arise in the textual modality $q_i$. To compute $f(y_i|q_i^s, v_i)$, we mask $q_i^c$ in the query before feeding it as input to $f(.)$.

## 6 Experiment Setup

**Datasets.** We evaluate the performance of our methods in both in-distribution (ID) and out-of-distribution (OOD) settings on multiple multimodal tasks, including VQA-CP (Agrawal et al., 2018b), GQA (Hudson and Manning, 2019), GQA-OOD (Kervadec et al., 2021), and NLVR2 (Suhr et al., 2019) datasets. To further assess robustness in the presence of language and vision biases, we create the **IVQA-CP** test set by replacing the original images in the VQA-CP test set with counterfactual images from IV-VQA (Agarwal et al., 2020). These IV-VQA images have been edited to remove irrelevant objects while preserving the original ground truth label (details in Appendix).

**Architectures.** We use the LXMERT (Tan and Bansal, 2019) model as our baseline and implement our methods TE-D and ATE-D on top of LXMERT for all datasets. Since VQA-CP is a reorganization of the VQA v2 dataset and LXMERT is pretrained on VQA v2, initializing pretrained LXMERT model for finetuning on VQA-CP leads to data leakage and an unreasonable increase in accuracy. Hence, we train LXMERT-based models, baselines from scratch in our experiments and are not comparable to numbers in Wen et al. (2021); Gokhale et al. (2020) affected by data leakage.

| | VQA-CP | | | | IVQA-CP | | | | Additional #MFLOPS |
|---|---|---|---|---|---|---|---|---|---|
| | Overall | Yes/No | Num | other | Overall | Yes/No | Num | other | |
| LXMERT (Tan and Bansal, 2019) | 41.2 | 44.1 | 13.9 | 47.2 | 35.0 | 43.3 | 12.7 | 36.8 | - |
| + IRM (Peyrard et al., 2022) | 42.7 | 44.1 | 15.2 | 49.5 | 36.5 | 43.2 | 12.8 | 39.3 | - |
| + ATE-D (ours) | 42.2 | 43.6 | 14.6 | 49.0 | 35.8 | 42.9 | 13.2 | 38.2 | **0.7** |
| + TE-D (ours) | 43.4 | 48.3 | 14.4 | 48.8 | 36.7 | 46.5 | 12.8 | 38.1 | 8.8 |
| + CD-VQA (Kolling et al., 2022b) | 42.1 | 42.7 | 14.8 | 49.3 | 36.3 | 44.7 | 12.9 | 38.7 | - |
| + GenB (Cho et al., 2023) | **52.8** | **67.3** | **29.8** | 49.7 | **41.3** | **50.7** | **16.7** | **39.4** | 50.2 |
| D-VQA$_f$ (Wen et al., 2021) | 43.9 | 47.5 | 15.7 | **49.8** | 37.3 | 45.8 | 13.9 | 39.2 | 18.9 |
| D-VQA$_f$ + ATE-D | 43.9 | 47.2 | **15.9** | 49.9 | 37.4 | 45.7 | 13.9 | 39.3 | 19.6 |
| D-VQA$_f$ + TE-D | **44.6** | **47.8** | 15.7 | **50.8** | **37.8** | **46.2** | 13.9 | **40.1** | 27.7 |
| D-VQA | 52.4 | 65.5 | 29.7 | 51.8 | 44.6 | 62.9 | 26.4 | 39.9 | 25.0 |

Table 1: Accuracy results on the VQA-CP (Agrawal et al., 2018a) and IVQA-CP (Agarwal et al., 2020) test sets. Higher is better. Column 'Additional MFLOPs' represents extra MFLOPS introduced by each method over the LXMERT backbone. We report results using a LXMERT model free of the data leakage issue.

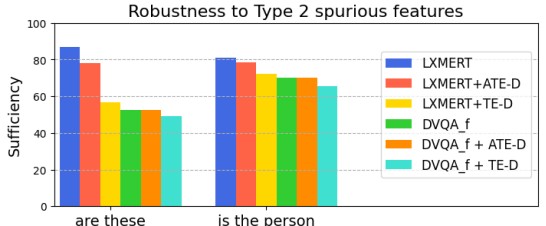

Figure 6: Using our sufficiency metric ($\lambda$, lower is better), we show that our debiased models rely less on Type 2 spurious features than baseline models.

# 7 Results & Discussion

In this section, we discuss the results from the evaluation of our methods for generalization, robustness, effectiveness, and efficiency, and analysis of the learned confounder representations.

## 7.1 Does causal debiasing help improve out-of-distribution generalization?

We evaluate the effect of causal debiasing on improving generalization by evaluating our methods on three multimodal datasets. First, we observe that our methods, ATE-D and TE-D, demonstrate 1% and 2.2% gains over LXMERT on the VQA-CP test set (see Tab. 1). TE-D improves the accuracy of Yes/No category by 4.2% which has higher bias presence as seen in Fig. 7 and outperforms D-VQA$_f$, a state-of-art unimodal debiasing method for VQA (feature perspective only), by 0.8% ($p$=0.04) [2] in the Yes/No category, while the latter achieves better overall accuracy on VQA-CP. However, our methods can be used to debias features in any backbone and task, in contrast to D-VQA$_f$ that has been designed for VQA. Moreover, D-VQA$_f$ trains a debiased model from scratch while TE-D debiases a biased model with a few epochs of fine-tuning (see efficiency in Sec. 7.4).

[2]Statistical significance is computed with 100K samples using bootstrap (Noreen, 1989; Tibshirani and Efron, 1993). All other gains are statistically significant.

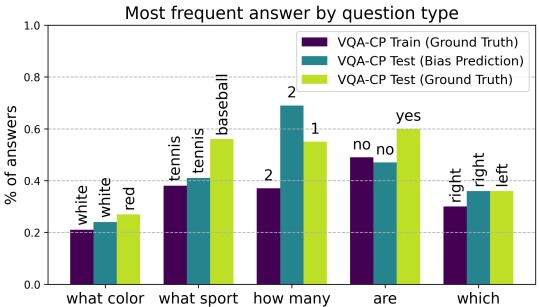

Figure 7: Most frequent answer by question type in VQA-CP train, test, and bias predictions from TE-D.

GenB (Cho et al., 2023) achieves state-of-the-art results on top of LXMERT by using ensembles of distilled models but compromises on efficiency. We see 1.8% and 2.3% gains in GQA-OOD accuracy with ATE-D and TE-D over the LXMERT baseline (see Tab. 2). The GQA-OOD dataset is further divided into OOD-Head and OOD-Tail splits which represent the samples containing answers from the head and tail of the answer distributions respectively; our methods achieve improvements in both groups. These gains are obtained along with gains in in-distribution (ID) accuracy on GQA (see Tab. 2). Additionally, we see 0.4%, 0.5% gains with ATE-D, TE-D respectively on NLVR2, an ID evaluation setting for visual entailment task (see Tab. 3). This shows that our methods do not hurt in-distribution performance and are task-agnostic.

## 7.2 What kind of biases are captured by confounder representations?

**ATE-D.** First, we find that up-weighting features similar to the confounders learned in ATE-D, as opposed to down-weighting (see Sec. 4.1), significantly hurts OOD accuracy implying that the confounder representations indeed encode spurious correlations. Next, we train a non-linear probe on the confounder representations for the VQA task. The accuracy of this probe is 25% and the distribu-

tion of predicted answers of this probe has lower entropy than that of the predicted answer distribution from unbiased features. Lower entropy suggests higher bias in the semantic concepts encoded in the confounders.

**TE-D.** The bias representations in TE-D capture the most prominent input-output biases in the VQA-CP train set, accounting for answers in 0.34% of the answer vocabulary but covering approximately 67% of the train questions. The classifier head connected to these bias representations achieves 28% accuracy on the VQA-CP test set, while the overall causal model accuracy is 44%. The most frequent answers predicted by this classifier head on the VQA-CP test set align with those in the VQA-CP train set, showing that the captured confounders effectively represent dataset biases (see Fig. 7).

### 7.3 Does causal debiasing improve robustness to spurious features?

**Type 1 Spurious Features.** In Sec. 5, we discuss Type 1 spurious features that are irrelevant to the target output. Our IVQA-CP test set (Sec. 6) shares question annotations with VQA-CP but has images edited to remove irrelevant objects (Agarwal et al., 2020). Models trained on VQA-CP are evaluated on this dataset, allowing assessment of their robustness to spurious features. The LXMERT baseline shows a significant drop from 41.2% to 35.0% on IVQA-CP (Tab. 1), indicating the evaluation's challenging nature. Our methods, ATE-D and TE-D, achieve 0.8% and 1.7% improvements respectively over LXMERT on the IVQA-CP test set, enhancing robustness to Type 1 features. D-VQA$_f$ performs explicit visual debiasing and hence, exhibits the highest robustness to Type 1 features in IVQA-CP.

**Type 2 Spurious Features.** A prominent source of Type 2 spurious features in VQA is the first few words of a question, as seen in Fig. 5. We introduce the sufficiency score ($\lambda$, see Eqn. 7) to understand whether debiasing methods truly improve the robustness of models to such spurious features. We select two question types i.e. questions starting with "Are these" and "Is this person", which are strongly biased in the training set of VQA-CP, and compute the sufficiency of the phrases for model predictions by masking the remaining question (see Sec 5). As shown in Fig. 6, we find that causal debiasing methods lower the sufficiency score of the spurious feature for both of these question types, suggesting that they indeed alleviate the reliance

| | GQA | GQA OOD | | |
| | ID | Tail | Head | All |
|---|---|---|---|---|
| LXMERT (Tan and Bansal, 2019) | 59.8 | 49.8 | 57.7 | 54.6 |
| +VILLA (Gan et al., 2020) | - | 49.9 | 57.2 | 54.5 |
| +MANGO (Li et al., 2020) | - | - | - | 54.9 |
| +X-CGM (Jiang et al., 2021) | - | 49.9 | 57.5 | 55.6 |
| +ATE-D (ours) | **60.0** | 50.8 | 59.9 | 56.4 |
| +TE-D (ours) | 59.9 | **51.4** | **60.1** | **56.8** |

Table 2: Accuracy results on GQA ID and OOD datasets for various debiasing methods. Higher is better.

| | Acc. | Cons. |
|---|---|---|
| LXMERT (Tan and Bansal, 2019) | 74.5 | 39.4 |
| +ATE-D (ours) | 74.9 | 39.9 |
| +TE-D (ours) | 75.0 | 39.6 |

Table 3: Accuracy (Acc.) and consistency (Cons.) results on NLVR2 ID test set. Higher is better.

of these models on spurious features for making predictions. TE-D and D-VQA$_f$ achieve similar sufficiency scores, suggesting that they are equally effective at improving robustness by giving more importance to the context. TE-D achieves lower $\lambda$ than ATE-D which aligns with its larger accuracy gains (see Tab. 1).

### 7.4 Is cross-modal debiasing more effective and efficient than unimodal debiasing?

D-VQA$_f$ outperforms cross-modal debiasing in Table 1, but when D-VQA$_f$ is treated as the biased model in TE-D, additional improvements of 0.7% ($p$=0.03) are achieved, indicating that cross-modal interactions contribute to bias not addressed by unimodal debiasing. Cross-modal feature-based confounders effectively mitigate biases involving multiple modalities. Our causal debiasing methods demonstrate higher efficiency compared to D-VQA, with ATE-D adding 0.7 MFLOPS and TE-D adding 3% additional parameters and 8.8 MFLOPS to LXMERT. In contrast, D-VQA adds 5% additional parameters and 18.9 MFLOPS during training, requiring more time as it is trained from scratch. Efficiency results for GQA and NLVR are the same as those reported for VQA.

## 8 Conclusion

We propose ATE-D and TE-D to mitigate biases in models by imposing causally-driven information loss on biased features to learn confounders. Experimental results across various multimodal tasks, datasets, and backbones demonstrate that the learned confounders capture biases successfully, and our methods effectively eliminate biases from both unimodal and multimodal interactions.

## 9 Limitations

While we evaluate robustness to spurious features, we do so on specific question types for Type 2 features and specific Type 1 features (irrelevant objects in the image). Getting an all-inclusive robustness metric for evaluating debiasing methods would be insightful. Approaches that debias using data augmentation or sample balancing, although cumbersome, are more effective than feature-based debiasing approaches, including those proposed in our paper. More analysis is required to understand how the merits of sample-perspective and feature-perspective methods can be merged efficiently.

## 10 Broader Impact

In this work, the biases that we try to mitigate stem from the spurious correlations present in the dataset that lead to a drop in performance in OOD settings. This helps models learn causal associations between inputs and targets and thus brings them closer to real-world deployment as it helps mitigate unethical use of these models. However, vision-language models may encode other societal stereotypes and biases present in the data they are trained on and also introduce new ones. VL models explored in this paper are not immune to these issues. We are hopeful that our focus on modeling biases and alleviating them is a step towards more inclusive models.

### Acknowledgments

We thank Peter Hase, Zhuofan Ying, and Jaemin Cho for their useful insights about this work, and the reviewers of this paper for their helpful feedback. This work was supported by ARO W911NF2110220, DARPA MCS N66001-19-2-4031, ONR N00014-23-1-2356, DARPA ECOLE Program No. HR00112390060. The views, opinions, and/or findings contained in this article are those of the authors and not of the funding agency.

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

# A Causal Theory Preliminaries

In this section, we discuss our proposed causal graph for multimodal tasks and the two causal mechanisms relevant to our debiasing methods.

**Causal Graph.** Causal graphs are directed acyclic graphs $\mathcal{G} = \{\mathcal{V}, \mathcal{E}\}$ where the edges $\mathcal{E}$ are used to represent causal relationships between random variables $\mathcal{V}$. An example is shown in Fig. 2(a), where $\mathbf{M}$ has a *direct effect* on $\mathbf{A}$. When the variable $\mathbf{Q}$ has an *indirect effect* on $\mathbf{A}$ through a variable $\mathbf{M}$ i.e. $\mathbf{Q} \rightarrow \mathbf{M} \rightarrow \mathbf{A}$, the variable $\mathbf{M}$ is said to be a *mediator* in the causal graph. If a variable $\mathbf{C}$ has a direct causal effect on both $\mathbf{M}$ and $\mathbf{A}$, it is said to be a *confounder*.

**Causal Perspective for Multimodal Tasks.** Models developed for multimodal tasks are designed to use the combined data stream of vision ($V$) and language ($Q$) for solving the task. However, the unimodal data variables may act as confounders and give rise to spurious features in the model e.g. via $Q \rightarrow M, Q \rightarrow A$. Existing approaches that leverage causal theory for debiasing multimodal models aim to eliminate the direct unimodal effects. However, consider the VQA example in Fig. 1. A potential spurious correlation that may lead to incorrect predictions from models on similar examples is that in most training instances where the question asks the color of an object, the object is present in the center of the image. Spurious correlations arising from such multimodal interactions are ignored in existing causal graphs for multimodal tasks. Hence, we propose to model the spurious correlation as a confounder $\mathbf{C}$ that affects the mediator $\mathbf{M}$ and the answer $\mathbf{A}$ (see Fig. 2(a)). This allows us to model the biases encoded in the multimodal features as confounder $\mathbf{C}$ and eliminate the bias using causal intervention.

In order to debias VQA models, we adopt two causal mechanisms i.e., the Average Treatment Effect (ATE) and Total Effect (TE), which essentially refer to the same effect but differ in how they deal with the confounder (VanderWeele, 2015; Tang et al., 2020a). In ATE, $C$ is treated as a distribution, and $c$ is sampled without assuming a causal association with the treatment $M = m$. In TE, $c$ is causally associated with the treatment $M = m$ in each sample. We explore both mechanisms in our experiments and discuss their theories below.

**Average Treatment Effect.** The aim of causal inference is to estimate the independent effect of an intervention on a treatment variable $M$ on an outcome of interest $A$ i.e. to estimate the conditional probability distribution $P(A|do(M))$. However, standard models are optimized to infer the observational conditional probability $P(A|M)$ and in the presence of confounders i.e. variables $c \in C$ that affect both $A$ and $M$

$$P(A|M) \neq P(A|do(M)) \qquad (8)$$

where the *do*-operation implies the causal effect of $M \rightarrow A$. $P(A|do(M))$ can be estimated using backdoor adjustment by controlling for all values of the confounders $c \in C$, i.e.,

$$P(A|do(M)) = E_{c\sim C}[P(A|M,c)] \qquad (9)$$

This translates to an empirical sum over all possible values of the confounder in practice, also known as average treatment effect (ATE) (see Fig. 2(b)). When the confounders are known and observed, the confounder values are selected using suitable rules and heuristics (Pearl et al., 2000).

**Total Effect.** We need to isolate the causal effect of $M = m$ on $A$, free from the influence of the confounders $C$. According to causal theory, the total effect (TE) of treatment $M = m$ on $A$ can be computed as,

$$TE = A_{m,C_m} - A_{m*,C_m} \qquad (10)$$

where $M = m*$ represents the "no treatment" condition and $C_m$ represents the confounder under the treatment condition i.e $M = m$. By retaining the confounder in both sides of the difference, we eliminate the direct effect of $C_m$ on $M$ (see Fig. 2(c)).

## A.1 ATE-D

Step-2 of ATE-D:

Inspired by feature reweighing (Kirichenko et al., 2022), we instantiate backdoor adjustment by re-calibrating $r_i$ based on confounder similarity i.e., $E_{\hat{c} \in D_{\hat{c}}}[f(R, \hat{c})]$ (see Fig. 2(b)) as,

$$P(A|do(Q), do(V)) = P(A|do(M)) \quad (11)$$

$$E_C[P(A|M, C)] = E_{\hat{c} \in D_{\hat{c}}}[P(A|M, \hat{c})] \quad (12)$$

$$\approx P(A|E_{\hat{c} \in D_{\hat{c}}}[f(M, \hat{c})]) \quad (13)$$

See appendix of Huang et al. (2022) for complete proof. In our analysis, we instantiate $f(.)$ as the cosine similarity function in $s(.)$, as discussed in Sec 4.1.

## B  Analysis

While OOD generalization accuracies are indicative of the model learning causal relationships between the inputs and labels, another way to probe causal learning is to investigate if the models are robust to spurious features present in the dataset. In order to evaluate this, in this section, we discuss an analysis framework for probing the behavior of models toward spurious features and propose a new metric for evaluation. Joshi et al. (2022) define the probability of necessity (PN) of a feature $X_i$ for predicting the label $Y$ as the probability that the ground truth label $Y$ changes when the feature $X_i$ is changed. Similarly, they define the probability of sufficiency (PS) of a feature $X_i$ for predicting the label $Y$ as the probability that setting $X_i = x_i$ in a sample where $X_i \neq x_i$ is absent changes its ground truth label $Y$. Based on this framework, spurious features are categorized into (a) *low PN, low PS features*: These features are irrelevant to the ground truth label e.g., person in the image when the VQA question is "How many trees are in the picture?" (see Fig. 5) (b) *High PN, low PS features*: These features are necessary but not sufficient to make predictions i.e. the model should rely on other features in their presence. For instance, when a model always answers "yes" to all questions starting with "Is the man.." irrespective of the image, the model is biased towards the feature "Is the man.." (see Fig. 5). Henceforth, we refer to the low PS, low PS, and high PN, low PS features as *Type 1* and *Type 2* features respectively. We use this framework to analyze the various debiasing methods in our experiments.

**Sufficiency.**  In order to evaluate the robustness to *sufficiency* of type 2 features, we propose a novel metric for quantifying the sufficiency of a feature towards a prediction. We define the certainty of predictions as the KL divergence between the predicted output distribution and uniform distribution across all samples in the group (Ying et al., 2022). We define the sufficiency score ($\lambda$) as the certainty of a model's prediction when only the non-spurious features are the input to the model. Further, in order to make this metric comparable across models, we normalize this with the certainty of the model's predictions when the complete sample i.e., spurious as well as non-spurious features, is the input to the model. This results in a metric that represents the percentage of certainty of the model that can be attributed to the non-spurious component of the input. For a data sample $(x, y)$, let the input $x$ be comprised of the spurious feature $x^s$ and the remaining context $x^c$ i.e. $x = [x^s; x^c]$. The sufficiency $\lambda$ is computed as follows:

$$\lambda = \frac{\sum_{i=1}^{G} \text{KL}(f(y_i|x_i^s)||\mathbf{U})}{\sum_{i=1}^{G} \text{KL}(f(y_i|x_i)||\mathbf{U})} \quad (14)$$

where $\mathbf{U}(.)$ represents the uniform distribution, $f(.)$ is the trained model, and $G$ is a group of samples. A good debiasing technique should increase the sufficiency of non-spurious features. For the multimodal VQA task where $x_i = (q_i, v_i)$, we focus on the type 2 features emerging in the text modality $q_i$. To compute $f(y_i|q_i^c, v_i)$, we mask $q_i^s$ in the query before sending it as input to $f(.)$.

## C  Experiment Setup

### C.1  Datasets

- VQA-CP (Agrawal et al., 2018a): It is a re-organization of the VQAv2 (Antol et al., 2015) such that the distribution of question type-answer correlation is different between the train and test splits. This evaluation helps demonstrate the method's ability to debias in a setting where language bias is dominant.

- VQA-CP + IV-VQA: We evaluate it on a new version of the VQA-CP test set where we replace the image in each sample with their invariant counterparts from the IV-VQA dataset from (Agarwal et al., 2020). IV-VQA dataset has images replaced with their edited version obtained after removing irrelevant objects in a way that the predicted answer does not change.

| Hyperparameter | LXMERT | ATE | TE |
|---|---|---|---|
| Learning Rate | 5e-5 | 5e-5 | 5e-5 |
| Epochs | 20 | 5 | 5 |
| Max Gradient Norm | 1.0 | 1.0 | 1.0 |
| Weight Decay | 0.0 | 0.01 | 0.01 |
| Batch Size | 32 | 32 | 32 |
| Max Length | 128 | 128 | 128 |
| Warmup Ratio | 0.1 | 0.1 | 0.1 |
| LR Decay | Linear | Linear | Linear |
| Optimizer | AdamW | AdamW | AdamW |
| Bias dimension factor | - | - | 4 |
| Confounder dictionary size | - | 10 | - |

Table 4: Training hyperparameters for different models trained on the VQA-CP dataset.

This adds another layer of hardness to the benchmark along the image dimension. This evaluation helps demonstrate the method's ability to debias in a setting where both language and vision biases are dominant.

- GQA(Hudson and Manning, 2019), GQA-OOD(Kervadec et al., 2021):
GQA evaluation helps measure visual reasoning as well as compositional question-answering abilities. GQA-OOD is a re-organization of the GQA dataset that introduces distribution shift in validation and test sets based on question type similar to VQA-CP.

- NLVR2 (Suhr et al., 2019): It helps the generalization to multimodal tasks other than question answering. It helps evaluate reasoning abilities about sets of objects, comparisons, and spatial relations.

All our experiments are run with a single seed value.

**Baselines.** We use D-VQA$_f$ (feature perspective only) (Wen et al., 2021) based on LXMERT as the baseline for experiments with VQA-CP and train from scratch due to the aforementioned reasons. We also present results from D-VQA (both feature & sample perspective) for comparison, however, note that methods using data balancing are not comparable to causal debiasing methods (see Sec. 1).

## D   Results

### D.1   Analysis of confounder features

We compare the most frequent answer in the VQA-CP training and test sets with those from the predictions of the bias classifier head in TE-D in Fig. 7. As discussed in Sec.5, the predictions from bias classifier head closely track the distribution of answers in VQA-CP training set, even though the VQA-CP test set distribution is significantly different from VQA-CP train. This shows that the confounder representations indeed capture the strong priors present in training set.

**Explanation and proof for biases stemming from multimodal interactions**   Multimodal models have been known to be brittle to linguistic biases [1] and visual biases [2]. In this work, we demonstrate the presence of multimodal biases and the need for removing those biases from multimodal features. (Proof) Many existing debiasing methods focus on removing each unimodal bias (e.g., linguistic) from multimodal features independently of the other unimodal biases (e.g., visual). However, [3] suggest that the biases can stem from multimodal interactions as well; they perform semantic edits on images in VQA (I-VQA dataset) that should not affect the ground truth, and show that the answers from multimodal models change in response to these invariant edits. (Existing Methods) Indeed, methods like D-VQA [2] leave large room for improvement in terms of performance on the IVQA-CP dataset [see Lines] that are designed to test for multimodal biases, as we show in Table 1. (Our Approach) We formalize this phenomenon through the causal graph proposed in our paper in Fig. 2, where we explicitly model the confounders

| | VQA-CP | | | | IVQA-CP | | | | Additional |
|---|---|---|---|---|---|---|---|---|---|
| | Overall | Yes/No | Num | other | Overall | Yes/No | Num | other | #MFLOPS |
| LXMERT (Tan and Bansal, 2019) | 41.2 | 44.1 | 13.9 | 47.2 | 35.0 | 43.3 | 12.7 | 36.8 | - |
| + IRM (Peyrard et al., 2022) | 42.7 | 44.1 | 15.2 | 49.5 | 36.5 | 43.2 | 12.8 | 39.3 | - |
| + ATE-D (ours) | 42.2 | 43.6 | 14.6 | 49.0 | 35.8 | 42.9 | 13.2 | 38.2 | **0.7** |
| + TE-D (ours) | 43.4 | **48.3** | 14.4 | 48.8 | 36.7 | **46.5** | 12.8 | 38.1 | 8.8 |
| D-VQA$_f$ (Wen et al., 2021) | **43.9** | 47.5 | **15.7** | **49.8** | **37.3** | 45.8 | **13.9** | **39.2** | 18.9 |
| D-VQA$_f$ + ATE-D | 43.9 | 47.2 | **15.9** | 49.9 | 37.4 | 45.7 | 13.9 | 39.3 | 19.6 |
| D-VQA$_f$ + TE-D | **44.6** | 47.8 | 15.7 | **50.8** | **37.8** | **46.2** | 13.9 | **40.1** | 27.7 |
| D-VQA | 52.4 | 65.5 | 29.7 | 51.8 | 44.6 | 62.9 | 26.4 | 39.9 | 25.0 |

Table 5: Accuracy results on the VQA-CP (Agrawal et al., 2018a)and IVQA-CP (Agarwal et al., 2020) test sets. Higher is better. Column 'Additional MFLOPs' represents extra MFLOPS introduced by each method over the LXMERT backbone. *We report results using a LXMERT model free of the data leakage issue.*

that affect the variable connecting multimodal representation (M) and the outcome (A). The unimodal biases are implicitly modeled via the multimodal variable (Q->M->A, V->M->A). (Example) We demonstrate an example of this phenomenon in Fig. 1, where D-VQA fails to answer a question from the IVQA-CP test set correctly, and our proposed method, TE-D, is able to answer correctly because of multimodal debiasing. (Empirical Results) Additionally, we show improvements on top of unimodal debiasing methods like DVQA(f) with our multimodal debiasing approach (see rows 6,7 in Table 1). Our goal in this work is to demonstrate the presence of multimodal biases and the need for multimodal debiasing along with the potential of confounder modeling via information loss in causal multimodal debiasing and our results support this claim.