# OpenReview forum: "Debiasing Multimodal Models via Causal Information Minimization"
_EMNLP/2023/Conference — EMNLP 2023 Findings_

### Official Review · Reviewer_JDbk · 2023-07-31

**Soundness:** 3

**Excitement:**

2: Mediocre: This paper makes marginal contributions (vs non-contemporaneous work), so I would rather not see it in the conference.

**Paper Topic And Main Contributions:**

This paper proposes a debiasing method for vision-language models in a multi-modality setting. The authors proposed two different methods for extracting features that are related to spurious correlation in the training data.

**Reasons To Accept:**

1. Two solutions from different angles.
2. Good visual cases to describe the content of this paper.

**Reasons To Reject:**

1. The line 262-266 is not easy to understand why there is a causal relationship? Why can the latent representation approximate possible observations?
2. It would be great to show some cases in explaining average treatment effect and total effect.
3. Line 282 is not clear or straightforward enough. In this paper, what bias is considered and why models over-rely on spurious features? Higher co-occurrence between some features and labels?
4. In the step 1 of section 4.1, why can leveraging the latent representation of a small autoencoder help to get the features related to spurious correlation? Intuitively smaller models are easy to learn spurious correlations but omit non-spurious correlations. However can we evaluate to what extent spurious correlation-related features have been extracted?
5. What is the format of g_j? Why equation (3)  has to be explained through backdoor adjustment?
6. The core assumption of the spurious feature extraction step in TE-D is spurious features should be of higher variance. Why?
7. What's the motivation behind ATE-D and TE-D in terms of average treatment effect and total effect respectively?
8. The related work about “Inductive Bias in Model Architecture” is not clear and does not show what typical inductive bias is like when it comes to the architect of a vision-language model. It would be great if the related work section can firstly show the full-picture of each related technique and then a brief description of each key work.

**Reproducibility:**

3: Could reproduce the results with some difficulty. The settings of parameters are underspecified or subjectively determined; the training/evaluation data are not widely available.

**Reviewer Confidence:**

2: Willing to defend my evaluation, but it is fairly likely that I missed some details, didn't understand some central points, or can't be sure about the novelty of the work.

---

> ### Author Rebuttal · Authors · 2023-08-29
>
> 1. **The line 262-266 is not easy to understand why there is a causal relationship? Why can the latent representation approximate possible observations?**: Confounders are variables that affect both the input and the outcome without creating a causal link between them (see path M->C->A in Fig 3). These undesirable correlations are caused by biases in the dataset which create a non-causal link between input and outcome (for example, the answer to most questions starting with “What sport” is “tennis” in the dataset. But “What sport” and “tennis” are not causally correlated). Past work [2] has shown that deep models preferentially encode dataset shortcuts under limited representation capacity and confounders can be modeled as low-dimensional vectors [Lines 179-182, 295-299]. We use the same principle in our proposed methods to learn confounders (or dataset biases) by imposing information loss on multimodal features. In ATE-D, the latent representations of an autoencoder are a result of compressing the original representations into low-dimensional vectors and are thereby expected to capture biases that represent the confounders [Lines 313-316, also see citations in Line 266]. Intuitively, confounders are biases which have low variance and high bias and feature compression helps capture biases.
>
>
>
> 2. **It would be great to show some cases in explaining average treatment effect and total effect.**: Since these are well-known theories in causality, it will be difficult to devote the limited space in the main text towards showcasing their examples, however, we will include some examples in Appendix and point the readers towards other appropriate resources on the internet like [1].
>
>
> 3. **Line 282 is not clear or straightforward enough. In this paper, what bias is considered and why models over-rely on spurious features? Higher co-occurrence between some features and labels?**: Yes, spurious features are the undesirable features that models tend to rely on because of higher correlations between certain features and the label in the dataset. The models tend to learn a causal relationship between the features and the label when actually it is not causal but caused by confounding variables that represent the spurious correlation. See Lines 66-72. Also see paper cited in line 282.
>
> 4. **In the step 1 of section 4.1, why can leveraging the latent representation of a small autoencoder help to get the features related to spurious correlation? Intuitively smaller models are easy to learn spurious correlations but omit non-spurious correlations. However can we evaluate to what extent spurious correlation-related features have been extracted?**: The latent representations are obtained by compressing the original features which means they capture the most common biases in the dataset which model the confounders. See a detailed explanation in our response to Question 1. We dedicate an entire subsection Section 7.2 to perform an analysis of whether the proposed methods actually learn the confounders that model biases in the dataset and show that the biases modeled by the learnt confounders indeed capture the dataset biases. See lines 530-554 and Figure 7.
>
>
> 5. **What is the format of $g_j$? Why equation (3) has to be explained through backdoor adjustment?**: It is an instantiation of backdoor adjustment which is commonly used for causal debiasing [1]. See Lines 243-266 in Section 3.
>
> 6. **The core assumption of the spurious feature extraction step in TE-D is spurious features should be of higher variance. Why?**: We assume that the reviewer meant lower variance, as mentioned in Lines 361-365. Spurious features are the biases present in the dataset that can be easily learned by a model with low modeling capacity (e.g., shallow neural network models) to ‘solve’ the dataset. These biases stem from high co-occurrence between inputs and outputs present in the dataset and represent simple explanations for why an input has a particular output. For example, in the VQA-CP dataset, most questions starting with ‘Is there’ have an answer ‘yes’. Thus, the features learnt by a shallow model will learn to only encode ‘Is the’ and ignore all of the following words in the question, resulting in low-information content and low variance.
>
>
> 7. **What's the motivation behind ATE-D and TE-D in terms of average treatment effect and total effect respectively?**: Both are debiasing methods that use the respective causal debiasing theories. See lines 243-280. We are instantiating the respective causal theories to debias with the help of confounders obtained by leveraging information minimization.
>
>
> 8. **The related work about “Inductive Bias in Model Architecture” is not clear and does not show what typical inductive bias is like when it comes to the architect of a vision-language model. It would be great if the related work section can firstly show the full-picture of each related technique and then a brief description of each key work.**: Thank you for the suggestion, we will follow your suggestion and reformat the Related Work with headers like this:
>
> - Unimodal inductive bias:  [3, 4, 5] studies utilize a separate QA branch in their models.  By doing so, the models become less dependent on language priors and are forced to focus more on visual information.
>
> - Confounder-modeling inductive bias:  Inductive bias of limiting the representational capacity of the model for encoding confounders in the data [6, 7].
>
> We will similarly reformat other parts of the related works section.
>
>
>
>
>
> [1] [Introduction to Causal Theory slides](https://www.bradyneal.com/slides/4%20-%20Causal%20Models.pdf)
>
> [2] Yang, Wanqian, et al. "Chroma-VAE: Mitigating Shortcut Learning with Generative Classifiers." Advances in Neural Information Processing Systems 35 (2022): 20351-20365.
>
> [3] Clark, Christopher, Mark Yatskar, and Luke Zettlemoyer. "Don’t Take the Easy Way Out: Ensemble Based Methods for Avoiding Known Dataset Biases." Proceedings of the 2019 Conference on Empirical Methods in Natural Language Processing and the 9th International Joint Conference on Natural Language Processing (EMNLP-IJCNLP). 2019.
>
> [4] Cadene, Remi, et al. "Rubi: Reducing unimodal biases for visual question answering." Advances in neural information processing systems 32 (2019).
>
> [5] Ramakrishnan, Sainandan, Aishwarya Agrawal, and Stefan Lee. "Overcoming language priors in visual question answering with adversarial regularization." Advances in Neural Information Processing Systems 31 (2018).
>
> [6] Kallus, Nathan, Xiaojie Mao, and Madeleine Udell. "Causal inference with noisy and missing covariates via matrix factorization." Advances in neural information processing systems 31 (2018).
>
> [7] Sen, Rajat, et al. "Contextual bandits with latent confounders: An nmf approach." Artificial Intelligence and Statistics. PMLR, 2017.

---

### Official Review · Reviewer_nFie · 2023-08-10

**Soundness:** 3

**Excitement:**

3: Ambivalent: It has merits (e.g., it reports state-of-the-art results, the idea is nice), but there are key weaknesses (e.g., it describes incremental work), and it can significantly benefit from another round of revision. However, I won't object to accepting it if my co-reviewers champion it.

**Paper Topic And Main Contributions:**

In this paper, the authors explore bias arising from confounding factors within a causal graph for multimodal data. The authors investigate a novel approach that employs information minimization based on causal motivations to acquire representations of these confounding factors and employ methods rooted in causal theory to mitigate bias within models. Our findings indicate that the acquired confounder representations effectively capture dataset biases, and the proposed debiasing techniques enhance the out-of-distribution (OOD) performance across multiple multimodal datasets, all while maintaining in-distribution performance. Moreover, The authors also identify the efficiency of spurious features in models' predictions, providing further evidence of the efficacy of our suggested approaches.


**Questions For The Authors:**

Question:

1. What’s the relationship between $C$ and $Q, A$? Are they independent?
2. I suggest the author to include more related works [1, 2] and comparing baselines [3, 4] on causal-based VQA debiasing.
3. The reference of VQA-CP in **Dataset** and Tab. 1 are **different**. Which dataset do you use?
4. (Follow-up) In the baseline method **D-VQA**, experiments are conducted on VQA-CP v2. Why not compare on the same dataset?
5. (Follow-up) Compared with the baseline method D-VQA (which is also trained with LXMERT), the proposed method (LXMERT+ATE-D or LXMERT+ATE-D) seems not achieve better performance. D-VQA_f +ATE-D still cannot outperform D-VQA. What are the advantages of the proposed method?
6. (To sum up) I hope the author can experiment on existing benchmarking and compare with more recent highly-related works to demonstrate their effectiveness.
7. In Tab. 1, why D-VQA_f + TE-D uses less #MFLOPS than D-VQA_f? I guess it needs two-stage training.
8. I suggest the author to provide some straightforward visualizations on how the proposed method is able to learn confounder.

[1] Niu, Yulei, et al. "Counterfactual vqa: A cause-effect look at language bias." *Proceedings of the IEEE/CVF Conference on Computer Vision and Pattern Recognition*. 2021.

[2] Niu, Yulei, and Hanwang Zhang. "Introspective distillation for robust question answering." *Advances in Neural Information Processing Systems* 34 (2021): 16292-16304.

[3] Kolling, Camila, et al. "Efficient counterfactual debiasing for visual question answering." *Proceedings of the IEEE/CVF winter conference on applications of computer vision*. 2022.

[4] Cho, Jae Won, et al. "Generative Bias for Robust Visual Question Answering." *Proceedings of the IEEE/CVF Conference on Computer Vision and Pattern Recognition*. 2023.

**Reasons To Accept:**

Strength:

1. This paper utilize causally-motivated information to learn confounder representations from biased features and utilize them to debias models. This approach offers a reasonable solution for making multi-modal models debiased.
2. The paper is well-structured and mostly clear.

**Reasons To Reject:**

Weakness:

1. In Fig. 1, when cup removed, I think “LXMERT blue” should be wrong.
2. My concerns mainly lie within the experimental section. See Questions.

**Reproducibility:**

4: Could mostly reproduce the results, but there may be some variation because of sample variance or minor variations in their interpretation of the protocol or method.

**Reviewer Confidence:**

4: Quite sure. I tried to check the important points carefully. It's unlikely, though conceivable, that I missed something that should affect my ratings.

---

> ### Author Rebuttal · Authors · 2023-08-29
>
> Response to Weaknesses:
>
> 1. **Error in Fig. 1**: Thanks for pointing it out. We’ll replace the word “blue” in first line of the second image with a “red”.
>
> Response to Questions:
>
> 1. **What’s the relationship between C, Q and A?**: As we mention in Lines 210-215, the variable C in the causal graph is the confounder that creates a non-causal link between the multimodal variable M (e.g., multimodal features in models like LXMERT) and the answer A. The question variable Q in the VQA task causally affects the multimodal variable M. Thus, C also creates a non-causal path between Q and A via M and C.(Q→ M→C←A). C is the confounder.
>
> 2. **Related works [1, 2] and baselines [3, 4] on causal-based VQA debiasing**: Thanks for suggesting the additional references. We implemented [3], [4] using a LXMERT model free of the data leakage issue we have explained in Lines 483-488 and the numbers are as follows:
>
> Method | Overall | Yes/No | Number | Other |
> --------------------------------------------- | ------- | ----- | ---- | --- |
> LXMERT(Tan and Bansal, 2019) | 41.2 | 44.1 | 13.9 | 47.2 |
> +CD-VQA [3] (*new*) | 42.1 | 42.7 | 14.8 | 49.3 |
> +ATE-D (ours) | 42.2 | 43.6 | 14.6 | 49.0 |
> +TE-D (ours) | 43.4 | 48.3 | 14.4 | 48.8 |
> +GenB [4] (*new*)| **52.8** | **67.3** | **29.8** | 49.7 |
> D-VQA*f* | 43.9 | 47.2 | 15.9 | 49.9 |
> D-VQA | 52.4 | 65.5 | 29.7 | **51.8** |
>
> Our methods outperform the data augmentation-based approach proposed in [3]. Especially, TE-D outperforms [3] by 1.3% without creating augmented data for balancing out the bias. Results show that [4]**\*** outperforms our methods as well as the previous SoTA i.e., D-VQA. [4] is an ensemble-based method and a multi-stage pipeline where they first train the (target) biased model, then train a separate generative model to model the bias and then use the generative model to debias the target model. Our work is different in that we model the bias from biased features using information theory and use it via causal methods to debias the target model in a single step. The goal of our work is not to achieve state-of-art results but to showcase the use of information minimization to learn confounders and the need for removing multimodal biases. We’ll add them to the Related Work section and the results in Table 1.
>
> **\*** The numbers in [4] has the same data leakage issue that we talk about in Lines 483-488. So their LXMERT numbers are misleading and not comparable to our results. The numbers that we report above are after removing this issue which is why they don’t match those reported in their paper [4].
>
> 3. **The reference of VQA-CP in Dataset and Tab. 1 are different. Which dataset do you use?**: Table 1 has Invariant VQA reference which we intended to put for the term IVQA-CP in the same line. The one in the dataset section is the actual  VQA-CP reference. We will correct it in the final version.
>
> 4. **(Follow-up) In the baseline method D-VQA, experiments are conducted on VQA-CP v2. Why not compare on the same dataset?**: When we say VQA-CP, we mean VQA-CP v2. We do not use VQA-CP v1 anywhere. Our numbers are not comparable to the numbers reported in the D-VQA paper because it has the data leakage issue that we avoid in our work. See Lines  483-488. We have reproduced D-VQA in a setting free of data-leakage and report those numbers in Table 1.
>
>
> 5. **(Follow-up) Compared with the baseline method D-VQA (which is also trained with LXMERT), the proposed method (LXMERT+ATE-D or LXMERT+ATE-D) seems not achieve better performance. D-VQA_f +ATE-D still cannot outperform D-VQA. What are the advantages of the proposed method?**: D-VQA explicitly models unimodal biases via unimodal branches, whereas we model biases using multimodal representations (See Figs 3, 4) without prior assumptions on what the biases look like which make it generalizable across modalities. Additionally, we demonstrate improvements over D-VQA_f with D-VQA_f + TE-D, showing that it is important to remove multimodal biases in addition to unimodal ones - which is one of the main goals of our work. D-VQA_f + ATE-D does not outperform D-VQA because D-VQA_f + ATE-D only performs feature-based debiasing whereas D-VQA is a combination of data augmentation and feature-based debiasing. We explicitly state that debiasing methods that use data augmentation are superior to debiasing methods that operate on features only [Lines 628-632]. Data augmentation is an expensive operation and needs prior knowledge about the kind of biases present in the dataset; hence, it is worth studying methods that perform feature-based debiasing only.
>
> 6. **(To sum up) I hope the author can experiment on existing benchmarking and compare with more recent highly-related works to demonstrate their effectiveness.**: Thanks! We have added the baselines in the response to Q2.
>
>
> 7. **In Tab. 1, why D-VQA_f + TE-D uses less #MFLOPS than D-VQA_f? I guess it needs two-stage training.**: We report the number of additional MFLOPS: D-VQA_f has 18.9 additional MFLOPS compared to LXMERT. D-VQA_f + TE-D has 8.8 additional MFLOPS compared to D-VQA_f. We will report the additional MFLOPS compared to LXMERT for D-VQA_f + TE-D to remove confusion in the updated version of our paper.
>
>
> 8. **I suggest the author provide some straightforward visualizations on how the proposed method is able to learn confounder.**: We dedicate subsection Sec 7.2 to performing an analysis of whether the proposed methods actually learn the confounders that model biases in the dataset and show that the biases modeled by the learnt confounders indeed capture the dataset biases. See lines 530-554 and Fig 7. Additionally, we will expand Figures 3 and 4 to show step-by-step breakdown of the model and emphasize on the confounder learning portion of the pipeline to remove any confusion.
>
> [1] Niu, Yulei, et al. "Counterfactual vqa: A cause-effect look at language bias." Proceedings of the IEEE/CVF Conference on Computer Vision and Pattern Recognition. 2021.
>
> [2] Niu, Yulei, and Hanwang Zhang. "Introspective distillation for robust question answering." Advances in Neural Information Processing Systems 34 (2021): 16292-16304.
>
> [3] Kolling, Camila, et al. "Efficient counterfactual debiasing for visual question answering." Proceedings of the IEEE/CVF winter conference on applications of computer vision. 2022.
>
> [4] Cho, Jae Won, et al. "Generative Bias for Robust Visual Question Answering." Proceedings of the IEEE/CVF Conference on Computer Vision and Pattern Recognition. 2023.

---

### Official Review · Reviewer_kR71 · 2023-08-12

**Soundness:** 2

**Excitement:**

3: Ambivalent: It has merits (e.g., it reports state-of-the-art results, the idea is nice), but there are key weaknesses (e.g., it describes incremental work), and it can significantly benefit from another round of revision. However, I won't object to accepting it if my co-reviewers champion it.

**Paper Topic And Main Contributions:**

This paper focuses on debiasing multimodal learning and proposes ATE-D and TE-D to mitigate biases in models by imposing causally-driven information loss on biased features to learn confounders. Experimental results across various multimodal tasks, datasets, and backbones demonstrate the superiority of the proposed method.

**Questions For The Authors:**

See weakness.

**Reasons To Accept:**

1. This paper is well-organized.
2. The authors propose two methods, TE-D and ATE-D, that leverage causally-motivated information loss to learn confounder representations from biased features.
3. The authors propose a sufficiency score to quantify the reliance of models on spurious features.

**Reasons To Reject:**

1. The authors point out that these existing approaches overlook biases stemming from multimodal interactions within their causal graphs. It is not convincing because it lacks explanations and proof.
2. The proposed problem and technologies are not novel in debiasing learning and multimodal learning, such as the causal graph, debiasing, the Average Treatment Effect, and the Total Effect.
3. There seems to be an error in Figure 1.
4. Why do the existing methods remove unimodal biases whereas the proposed method removes biases arising from cross-modal interactions? The authors should explain it by combining Figure 1.
5. Why variables can be used as substitutes for the confounders via latent representations in autoencoders?

**Reproducibility:**

4: Could mostly reproduce the results, but there may be some variation because of sample variance or minor variations in their interpretation of the protocol or method.

**Reviewer Confidence:**

5: Positive that my evaluation is correct. I read the paper very carefully and I am very familiar with related work.

---

> ### Author Rebuttal · Authors · 2023-08-29
>
> We thank the reviewer for their feedback. Please see our response below:
>
> - **Explanation and proof for biases stemming from multimodal interactions**: Multimodal models have been known to be brittle to linguistic biases [1] and visual biases [2]. In this work, we demonstrate the presence of multimodal biases and the need for removing those biases from multimodal features. **(Proof)** Many existing debiasing methods focus on removing each unimodal bias (e.g., linguistic) from multimodal features independently of the other unimodal biases (e.g., visual). However, [3] suggest that the biases can stem from multimodal interactions as well; they perform semantic edits on images in VQA (I-VQA dataset) that should not affect the ground truth, and show that the answers from multimodal models change in response to these invariant edits. **(Existing Methods)** Indeed, methods like D-VQA [2] leave large room for improvement in terms of performance on the IVQA-CP dataset [see Lines] that are designed to test for multimodal biases, as we show in Table 1. **(Our Approach)** We formalize this phenomenon through the causal graph proposed in our paper in Fig. 2, where we explicitly model the confounders that affect the variable connecting multimodal representation (M) and the outcome (A). The unimodal biases are implicitly modeled via the multimodal variable (Q->M->A, V->M->A). **(Example)** We demonstrate an example of this phenomenon in Fig. 1, where D-VQA fails to answer a question from the IVQA-CP test set correctly, and our proposed method, TE-D, is able to answer correctly because of multimodal debiasing. **(Empirical Results)** Additionally, we show improvements on top of unimodal debiasing methods like DVQA(f) with our multimodal debiasing approach (see rows 6,7 in Table 1). Our goal in this work is to demonstrate the presence of multimodal biases and the need for multimodal debiasing along with the potential of confounder modeling via information loss in causal multimodal debiasing and our results support this claim. We will emphasize this explanation in the Introduction section of our paper in its updated version.
>
>
> - **The proposed problem and technologies are not novel in debiasing learning and multimodal learning, such as the causal graph, debiasing, the Average Treatment Effect, and the Total Effect**: Yes, the task of debiasing in multimodal learning is not novel. In fact, it is a widely studied problem with significant implications for real-world deployment of such models. Moreover, technologies like causal graphs, Average Treatment Effect, Total effect are core tenets of causal theory and have been widely used in debiasing papers that leverage causal theory [6,7]. We do not make any claims about the novelty of the task or the aforementioned technologies, and accordingly, we place them under the section named ‘Preliminaries’. Our paper is motivated by the importance of the debiasing task and the simplicity of the causal framework, and our method addresses limitations of existing debiasing methods. Our novelty lies in the fact that we use information theory to learn confounders and we remove biases caused by multimodal interactions (Lines 139-142) by instantiating causal debiasing methods. To our knowledge, we are the first to propose this approach.
>
>
>
> - **Error in Figure 1**: Thanks for pointing it out. We’ll replace the word “blue” in the first line of the second image with a “red”.
>
>
> - **Why do the existing methods remove unimodal biases whereas the proposed method removes biases arising from cross-modal interactions? The authors should explain it by combining Figure 1**: Thank you for the suggestion. Please see our answer to question 1 as a response to this question. Let us know if our answer requires further clarification.
>
> - **Why variables can be used as substitutes for the confounders via latent representations in autoencoders?**: Confounders are variables that affect both the input and the outcome without creating a causal link between them [4] (see path M->C->A in Fig 3). These undesirable correlations are caused by biases in the dataset which create a non-causal link between input and outcome (for example, the answer to most questions starting with “What sport” is “tennis” in the dataset. But “What sport” and “tennis” are not causally correlated). Past work [5] has shown that deep models preferentially encode dataset shortcuts under limited representation capacity and confounders can be modeled as low-dimensional vectors [Lines 179-182, 295-299]. We use this principle in our proposed methods to learn confounders (or dataset biases) by imposing information loss on multimodal features. In ATE-D, the latent representations of an autoencoder are a result of compressing the original representations into low-dimensional vectors and are thereby expected to capture biases that represent the confounders [Lines 313-316]. Intuitively, confounders are biases which have low variance and high bias and feature compression helps capture biases.
>
>
> [1] Goyal, Yash, et al. "Making the v in vqa matter: Elevating the role of image understanding in visual question answering." Proceedings of the IEEE conference on computer vision and pattern recognition. 2017.
>
> [2] Wen, Zhiquan, et al. "Debiased visual question answering from feature and sample perspectives." Advances in Neural Information Processing Systems 34 (2021): 3784-3796.
>
> [3] Agarwal, Vedika, Rakshith Shetty, and Mario Fritz. "Towards causal vqa: Revealing and reducing spurious correlations by invariant and covariant semantic editing." Proceedings of the IEEE/CVF Conference on Computer Vision and Pattern Recognition. 2020.
>
> [4] VanderWeele, Tyler J., and Ilya Shpitser. "On the definition of a confounder." Annals of statistics 41.1 (2013): 196.
>
> [5] Yang, Wanqian, et al. "Chroma-VAE: Mitigating Shortcut Learning with Generative Classifiers." Advances in Neural Information Processing Systems 35 (2022): 20351-20365.
>
> [6] Mohammad Taha Bahadori and David Heckerman. 2020. Debiasing concept-based explanations with causal analysis.In International Conference on Learning Representations
>
> [7] Wenkai Zhang,Hongyu Lin,Xianpei Han,and Le Sun. 2021. De-biasing distantly supervised named entity recognition via causal intervention. In Proceedings of the 59th Annual Meeting of the Association for Computational Linguistics and the 11th International Joint Conference on Natural Language Processing (Volume1: Long Papers), pages 4803–4813, Online. Association for Computational Linguistics.

---

### Meta-Review · Area_Chair_KTMm · 2023-09-19

**Recommendation:** 4

**Metareview:**

The paper presents an approach for debiasing VQA models by identifying features that are likely to have spurious correlations with labels---those that are have low information content but are still predictive of the labels---using methods motivated by causal theory (Average Treatment Effect and Rate Distortion), and mitigate those correlations by feature reweighting and additional losses. Experiments compare the proposed approaches with models that do not use any debiasing or use debiasing techniques leveraging knowledge of biases, including unimodal feature based debiasing and data augmentation. The proposed approaches do better than all these except those that use data augmentation, but the authors argue that data augmentation can often be expensive and requires knowledge of biases.

The reviewers recommended providing additional explanations and visualizations to further clarify how the methods work and the authors have agreed to do so.

In addition, I would encourage the authors to include
1. a discussion on the limitations of targeted data augmentation methods for debiasing models, and in what scenarios _discovering_ spurious correlations would be practically more useful. It might help to include experiments on a wider range of distribution shifts to show the benefits of the proposed techniques over data augmentation. This would be helpful particularly given the large gap in performance between D-VQA and the proposed method.
2. a discussion on how the proposed approach relates to work on slice discovery methods (e.g.: Eyuboglu et al., 2022).


Reference

Eyuboglu et al., 2022: Domino: Discovering Systematic Errors with Cross-Modal Embeddings, ICLR 2022

---

### Decision · Program_Chairs · 2023-10-07

**Decision:**

Accept-Findings

**Comment:**

The paper presents an approach for debiasing VQA models by identifying features that are likely to have spurious correlations with labels---those that are have low information content but are still predictive of the labels---using methods motivated by causal theory (Average Treatment Effect and Rate Distortion), and mitigate those correlations by feature reweighting and additional losses. Experiments compare the proposed approaches with models that do not use any debiasing or use debiasing techniques leveraging knowledge of biases, including unimodal feature based debiasing and data augmentation. The proposed approaches do better than all these except those that use data augmentation, but the authors argue that data augmentation can often be expensive and requires knowledge of biases.

The reviewers recommended providing additional explanations and visualizations to further clarify how the methods work and the authors have agreed to do so.

In addition, I would encourage the authors to include
1. a discussion on the limitations of targeted data augmentation methods for debiasing models, and in what scenarios _discovering_ spurious correlations would be practically more useful. It might help to include experiments on a wider range of distribution shifts to show the benefits of the proposed techniques over data augmentation. This would be helpful particularly given the large gap in performance between D-VQA and the proposed method.
2. a discussion on how the proposed approach relates to work on slice discovery methods (e.g.: Eyuboglu et al., 2022).


Reference

Eyuboglu et al., 2022: Domino: Discovering Systematic Errors with Cross-Modal Embeddings, ICLR 2022